# Different effects of plasmids harboring $bla_{OXA-232}$ between major and minor clones in *Klebsiella pneumoniae*

Yun Young Cho,[1] Hyun Woo Kim,[2] Si Ho Kim,[3] Yu Mi Wi,[3] Kwan Soo Ko[1]

**ABSTRACT** This study investigated the differing impact of introducing the globally reported $bla_{OXA-232}$-harboring ColE-type plasmid on the plasmid stability, competitiveness, virulence, and gene expression of *Klebsiella pneumoniae* transconjugants from major high-risk genotypes (ST11, ST15, and ST307) compared to minor genotypes. Plasmid transformation was performed with electroporation. The carbapenem resistance, plasmid stability, competitive fitness, and virulence were assessed through antibiotic susceptibility profiling, plasmid stability, competition assay, human serum resistance assay, and quantitative real-time PCR. Consistent plasmid stability was evident across all genotypes and nutrient conditions tested following the introduction of the plasmid. *K. pneumoniae* transconjugants from major clones demonstrated increased competitiveness and virulence compared to their wild-type counterparts, while those from minor clones did not. Increased expression of the virulence-associated *fimH* gene was observed only in the transconjugants from the major clones. These results highlight a complex interplay between plasmids and bacterial hosts and explain why specific clones producing carbapenemase are frequently observed globally.

**IMPORTANCE** OXA-232 is a variant of OXA-48-like carbapenemase, exhibiting high hydrolytic activity against penicillin and weak activity against carbapenems. It was associated with widespread dissemination of carbapenem-resistant Enterobacteriaceae including *Klebsiella pneumoniae*. Particularly, it has been identified that high-risk clones (ST11 and ST15) of *K. pneumoniae* frequently harbor the plasmid with $bla_{OXA-232}$, that is, ColE-type plasmid. In this study, it is revealed that *K. pneumoniae* of high-risk clones with ColE-type plasmid harboring $bla_{OXA-232}$ showed an increase in competitiveness and virulence. It may explain why certain carbapenemase-producing clones are widespread worldwide, emphasizing the need to focus on coping with antibiotic resistance.

**KEYWORDS** $bla_{OXA-232}$, carbapenem resistance, virulence, plasmid stability, competitive fitness

**Peer Reviewers** Jun-Seob Kim, Incheon National University, Incheon, South Korea; Fernando Hernandez Quiñones, Universidad Autonoma de Chihuahua Facultad de Medicina, Chihuahua, Mexico

Address correspondence to Kwan Soo Ko, ksko@skku.edu.

The authors declare no conflict of interest.

See the funding table on p. 7.

The growing resistance to carbapenem antibiotics is concerning given their crucial role in clinical settings. Carbapenems serve both as empirical therapy and last-resort treatment. With their broad spectrum, carbapenems exhibit remarkable stability against various beta-lactamases (1). Resistance to carbapenems is due primarily to carbapenemases, a diverse group of enzymes capable of hydrolyzing carbapenems, penicillins, cephalosporins, and monobactams (2). Carbapenemases can be categorized into three classes (A, B, and D), with KPC-, NDM-, IMP-, VIM-, and OXA-48-like types being the most prevalent (3).

The OXA-232 enzyme is a variant of OXA-48-like carbapenemases belonging to class D. It exhibits higher hydrolytic activity against penicillin and weaker activity against carbapenems (4). Despite limited carbapenem resistance, OXA-232 is associated with widespread dissemination within various species of the Enterobacteriaceae family (5).

High-risk clones of *Klebsiella pneumoniae*, such as ST11 and ST15, harbor the plasmid with $bla_{OXA-232}$ (6).

The spread of carbapenemase-producing Enterobacteriaceae often involves plasmids, with various incompatibility types, such as IncF, IncX3, and ColE, carrying specific carbapenemase genes alongside virulence genes (7). Certain carbapenemases have been linked to particular hosts. For example, the $bla_{KPC-2}$ gene, which is often carried within IncF plasmids, is associated with *K. pneumoniae* CC258 and CC307 (8). $bla_{OXA-232}$ is frequently carried by ColE-type plasmids within *K. pneumoniae* high-risk clones (9). As their existence is often considered conceptually inexplicable, the widespread distribution of plasmids in bacterial populations is often referred to as the "plasmid paradox" (10).

In this study, a ColE-type plasmid was introduced into clinically isolated carbapenem-susceptible *K. pneumoniae* isolates of various genotypes, and the variations in the isolates with the plasmid were examined with respect to plasmid stability, competitive fitness, and virulence. Our findings explain the dissemination of certain carbapenemase-producing clones and emphasize the intricate interplay between plasmids and bacterial hosts in the context of fitness and virulence.

## MATERIALS AND METHODS

### Bacterial isolates

Thirty-nine clinical isolates of *K. pneumoniae* susceptible to carbapenems (designated as KCS01 through KCS41 but excluding KCS20 and KCS22) were obtained from the Samsung Changwon Hospital (Changwon, South Korea) between 2008 and 2022 (Table S1). Initial species identification was carried out by VITEK-2 (bioMerieux, France) and confirmed through 16S rRNA gene sequencing. Genotypes were determined using multi-locus sequence typing based on primers specified in Diancourt et al. (11). The isolates were categorized into major clones (ST11, ST15, and ST307) or minor clones (ST23, ST105, ST165, ST298, ST355, ST356, ST358, ST365, and ST469) as described in Wyres and Holt (12) and Spadar et al. (13). While multiple isolates for each genotype could be included for major clones, only a single isolate for each genotype could be included for minor clones. It may be one of the limitations in our study, but we tried to supplement the limitation by including several genotypes of minor clone. The presence of additional carbapenemase genes was examined via polymerase chain reaction (PCR) amplification using the primers listed in Table S2.

### Plasmids

Samples of the plasmid pM5_OXA, which harbors the carbapenemase gene $bla_{OXA-232}$; replicase, a hypothetical gene; a mobile gene cassette (MOB module); and the partial noncoding genes *ereA* (for erythromycin resistance) and *vbhA* (for partial antitoxin) were obtained from a previous study (GenBank accession number CP031737; Fig. S1) (14) and were identified as ColE-type plasmids. The pM5_OXA was introduced into all 39 *K. pneumoniae* clinical isolates via electroporation. The resulting transconjugants were similar to KCS01/pM5_OXA.

### Antibiotic susceptibility testing

Minimum inhibitory concentrations (MICs) were measured by broth microdilution following Clinical and Laboratory Standards Institute criteria (15) for the seven antibiotics ampicillin, imipenem, meropenem, cefotaxime, ciprofloxacin, amikacin, and gentamicin. *Escherichia coli* ATCC 25922 and *Pseudomonas aeruginosa* ATCC 27853 were used as controls for respective duplicated tests.

### Plasmid stability

To measure plasmid retention frequency, a plasmid stability assay was conducted for all isolates, as described by Lee et al. (14, 16) with slight modifications. Seed cultures

were obtained for each transconjugant, intact, multiple, double, and single gene-deleted plasmid by culturing on selective Luria–Bertani (LB) media. These seed cultures were inoculated in equal volumes and concentrations in nonselective broth media and allowed to grow overnight. Subculturing of each sample continued for 12 days in LB media for all tested transconjugants, with the additional use of M9 media exclusively for the intact plasmid transconjugants. The presence of the plasmids was confirmed through daily broth microdilution and spot tests on both LB and meropenem (2 mg/L) growth media. The presence of the plasmids was confirmed once every 2 days for two colonies from selective media for each plasmid transconjugant via PCR using the primers listed in Table S2.

## Competition assay

An *in vitro* competition assay was conducted using MG1655 as control and overnight cultures of both wild-type and transconjugants for 10 clinical isolates: seven isolates of major STs (three of ST11 and two each of ST15 and ST307) and three isolates of minor STs (one each of ST27, ST298, and ST469) (Table S1). These cultures were inoculated in phosphate-buffered saline (PBS) to achieve a 0.5 McFarland standard, MG1655, and each of the wild-type and transconjugants was inoculated individually at a 1:1 ratio in 10 mL of LB broth and incubated at 37°C for 24 h with shaking. Samples were collected at 0 h and after 24 h of incubation. The survival rate was determined by calculating the competition index (CI) with the obtained colony-forming units (CFUs) through serial 10-fold dilutions and spot testing on LB agar plates with meropenem concentrations of 32 mg/mL and 2 mg/mL at time 0 h and after 24 h, respectively. The CI was used as defined in Lee et al. (14).

## Human serum resistance assay

Serum resistance assays were carried out on 10 isolates of *K. pneumoniae* as used in the *in vitro* competition assay, along with their intact transconjugants, following a previously described protocol[19] with slight modifications. In summary, bacterial cultures grown to mid-log phase (with an optical density of 600 nm of 0.5) were washed in 1 mL aliquots and resuspended in PBS at a ratio of 1:100. Subsequently, 25 µL of the washed and resuspended bacterial cultures were exposed to normal human serum (NHS), obtained from Innovative Research (Novi, MI, USA). As a control, bacterial culture treated with heat-inactivated human serum (HIS) was used to determine the bactericidal effect of NHS. All samples were incubated with shaking for 3 h at 37°C. Following incubation, the samples were serially diluted with PBS and plated on blood agar. The number of CFUs surviving after NHS treatment was compared with the number surviving after treatment with HIS. All assays were conducted in triplicate, and the results are expressed as survival percentages.

## Gene expression

Quantitative real-time PCR (qRT-PCR) was conducted to compare the relative mRNA expression levels of the *fimH* gene in 10 wild-type *K. pneumoniae* isolates and their transconjugants. Total RNA was extracted from cultures of the isolates grown to mid-log phase using the RNeasy mini kit (Qiagen, Hilden, Germany). Subsequently, qRT-PCR was performed with a QuantStudio 6 Flex Real-Time PCR system (Applied Biosystems, Carlsbad, CA), utilizing TB Green Premix Ex Taq (TaKaRa, Shiga, Japan) and the primers specified in Table S2. The mRNA expression level of each target gene was normalized to the expression level of the housekeeping gene *rpoB* using the ΔΔCT method. All assessments were carried out in triplicate for accuracy and consistency.

## Statistical analyses

Statistical analyses were completed using Student's *t*-test in Prism version 8.3.0 software for Windows (GraphPad Software, San Diego, CA). Statistical significance was defined as *P* <0.05, *P* <0.01, or *P* <0.001.

## RESULTS

### Carbapenem resistance in transconjugants with pM5_OXA

All transconjugants of 39 *K. pneumoniae* isolates with the pM5_OXA resulted in an increase in carbapenem MICs (Table S1). While the carbapenem MICs of most transconjugants ranged from 2 to 8 mg/L, the MICs of three transconjugants of ST11 isolates against both imipenem and meropenem were higher. Two transconjugants, KCS19/pM5_OXA and KCS21/pM5_OXA, exhibited carbapenem MIC profiles of 16 mg/L, while KCS08/pM5_OXA showed a particularly high MIC of 64 mg/L (Table S1).

### Plasmid stability

Observations of plasmid stability revealed that pM5_OXA achieved high stability irrespective of nutritional conditions (LB and M9 media) for all transconjugants regardless of genotype (see Fig. 1A and B). All intact plasmid transconjugants maintained the plasmid state for 12 days in rich nutrients (LB media) and for 7 days in a nutrient-deficient condition (M9 media).

### Competitiveness

The competitive fitness of the 10 transconjugants was measured through an *in vitro* competition assay against *E. coli* MG1655 and then compared with that of their wild-type counterparts. A CI greater than one implies that the transconjugant is more competitive than its wild-type isolate. All but one transconjugant of isolates belonging to major STs had CI values significantly higher than one, indicating an increase *in vitro* competitiveness following the introduction of the plasmid (see Fig. 2A). However, the CI values of transconjugants of minor STs did not deviate from one. Overall, transconjugants of major STs were more competitive than their wild-type, susceptible recipients (see Fig. 2B and C).

### Serum resistance

An *in vitro* human serum assay also revealed differences in the effects of plasmid pM5_OXA between major and minor clones (see Fig. 3A and B). The survival rates of

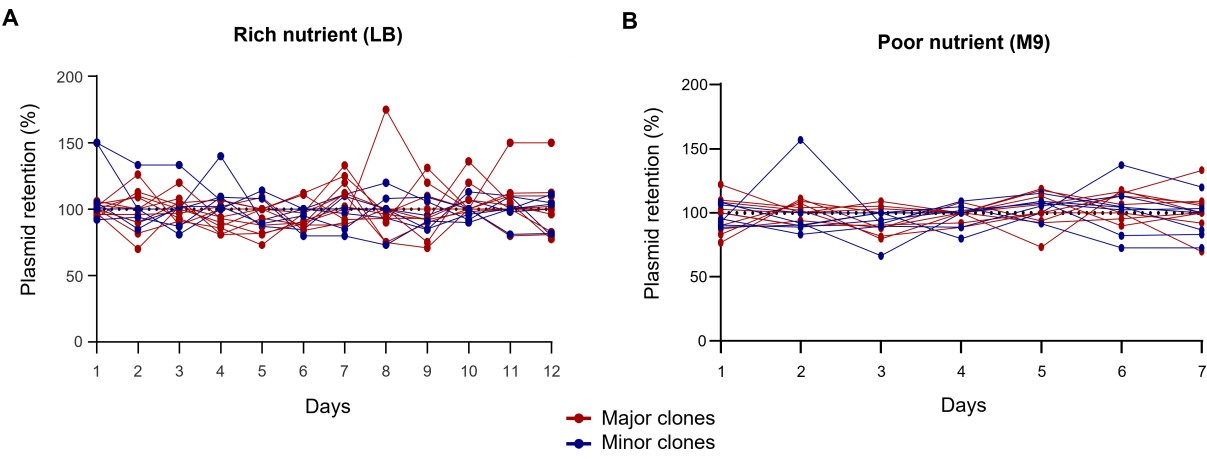

**FIG 1** Plasmid stability. The plasmid retention rate in (A) a rich nutrient condition (LB media) for 12 days and (B) a poor nutrient condition (M9 media) for 7 days. pM5_OXA was stable within a host irrespective of nutrient conditions for all transconjugants of genotype.

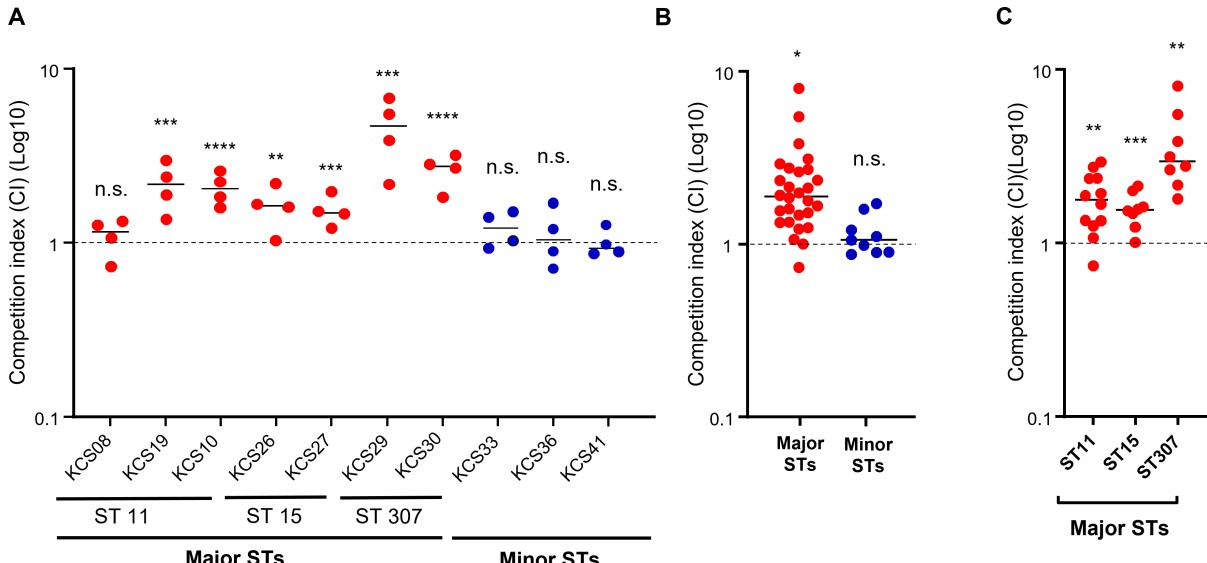

**FIG 2** Competition assay results of transconjugants against *E. coli* MG1655. (A) The competition index of 10 individual isolates, seven isolates of major clones (ST11, ST15, and ST307), and three isolates of minor clones. (B) A comparison of competition index between major clones and minor clones. (C) The competition index of three major clones. Statistical significance was evaluated against the neutral fitness (CI value = 1). *, $P < 0.5$; **, $P < 0.01$; ***, $P < 0.001$; ****, $P < 0.0001$; n.s., not significant.

the 10 clinical *K. pneumoniae* isolates against human serum were diverse, which was not consistent with their genotypes. However, all but one transconjugant of major clones (ST11, ST15, and ST307) showed significantly increased survival rates against human serum compared with their parental wild-type isolates. The survival rate against human serum was higher in one transconjugant from KCS26 of ST15, but the difference was not significant. On the other hand, all three transconjugants originated from isolates of minor STs, KCS33/pM5_OXA, KCS35/pM5_OXA, and KCS39/pM5_OXA, showed no difference in survival rates against human serum compared with those of their parental wild-type isolates, KCS33, KCS35, and KCS39, respectively.

### *fimH* gene expression

The *fimH* gene is associated with virulence as it expresses an adhesive subunit of type one fimbriae in *K. pneumoniae* (16). We compared the expression of *fimH* genes between wild-type carbapenem-susceptible isolates and their carbapenem-resistant transconjugants with pM5_OXA (Fig. 3C). The results were consistent with those of a serum resistance assay: most transconjugants of major clones showed significantly higher *fimH* gene expression compared with their corresponding wild types. A single transconjugant (KCS26/pM5_OXA) did not exhibit increased survival against human serum. For transconjugants from minor clones, one (KCS30/pM5_OXA) showed increased *fimH* expression in transconjugants, while decreased expression was evident in the other two.

### DISCUSSION

Plasmids may interact differently depending on the bacterial host, and this can influence their prevalence and survival (10). Different host-plasmid interactions may contribute to the dissemination or dominance of particular antibiotic-resistant clones, including carbapenemase-producing Enterobacteriaceae. Among carbapenemase-producing *K. pneumoniae* isolates, ST11, ST15, and ST307 are prevalent worldwide (16). $bla_{OXA-232}$, which is a variant of $bla_{OXA-48}$, is also frequent in the high-risk clones. In this study, the effects of $bla_{OXA-232}$-harboring plasmids were investigated among clones of high-risk genotypes (ST11, ST15, and ST307) and minor genotypes in *K. pneumoniae*.

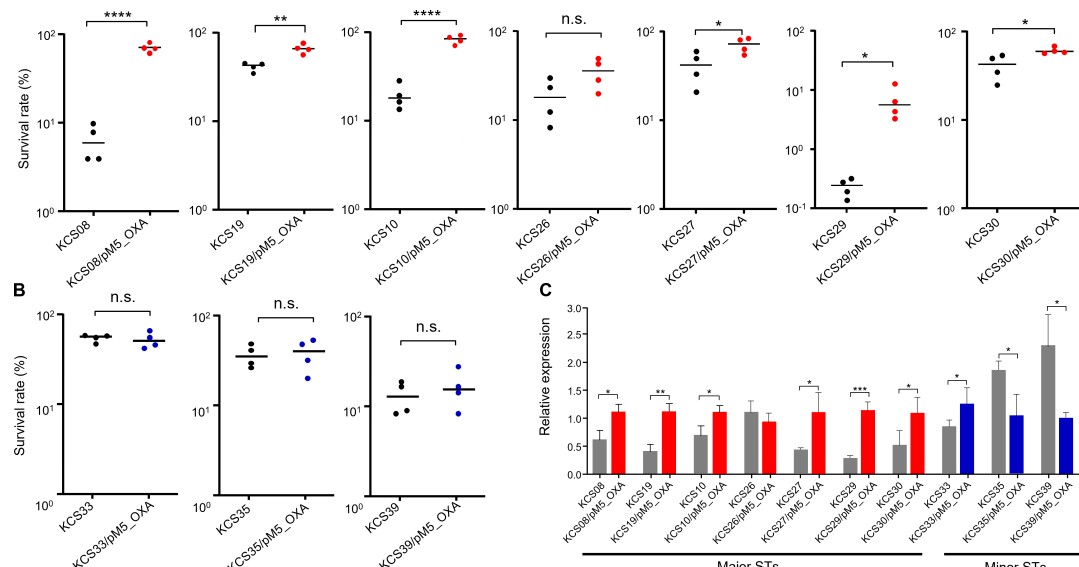

**FIG 3** Comparison of survival rates against human serum and *fimH* gene expression between susceptible wild-type recipients and their transconjugants with pM5_OXA. (A) The results for seven isolates of major clones (ST11, ST15, and ST307). (B) The human serum survival results in three isolates of minor clones. (C) *fimH* gene expression differences between susceptible wild-type recipients and their transconjugants with pM5_OXA. *, $P <0.5$; **, $P <0.01$; ***, $P <0.001$, ****, $P <0.0001$; n.s., not significant.

A ColE-type plasmid harboring *bla*$_{OXA-232}$ was well maintained in all *K. pneumoniae* isolates despite genotype and was stable in conditions of both rich and poor nutrients. The conjugation frequencies did not differ among the bacterial hosts. The stability of the ColE-type plasmid, even in *E. coli*, was identified previously (17). Plasmid stability may not affect the distribution of OXA-232-producing *K. pneumoniae* clones.

In contrast to plasmid stability within bacterial hosts, the effects of plasmids on *in vitro* competitiveness and survival against human serum differed between major and minor clones. Plasmid introduction significantly increased fitness and survivability against serum in high-risk clones but not in minor clones. In addition, *fimH*, which is associated with virulence (18), was overexpressed in *K. pneumoniae* isolates of major clones following the introduction of the plasmid. Plasmids tend to impose fitness burdens on their bacterial hosts. However, many studies have shown that plasmids can confer advantages even in the absence of positive selection, including antibiotics, which is termed the "plasmid paradox" (14). Our results show that this paradox is more evident in bacterial isolates of particular clones.

Our results explain in part why particular antibiotic-resistant clones are disseminated worldwide. The disseminated clones may show increased competitiveness and virulence through uptake of plasmids with a resistance gene. A recent study found that the introduction of plasmids with *bla*$_{OXA-48-like}$ genes may confer beneficial effects on some bacterial isolates (19). As a representative of plasmid with *bla*$_{OXA-48-like}$ genes, it has been known that ColE-type plasmid plays a role in killing other bacteria as well as antibiotic resistance (20). Bacterial isolates with advantages by plasmid were associated with phylogenetic grouping, and ST11 isolates showed beneficial fitness effects. It has been proposed that the fitness effects from plasmids are influenced by the accessory genome, including mobile genetic elements (21).

Because the ColE-type plasmid is small, at only 6,141 base pairs, it in itself seems to impose little in the way of fitness burden (14). However, it stimulates certain genetic pathways of a host bacterium that belongs to a particular clone, and this can have a positive effect on bacterial survival. The increase in virulence shown in this study may be associated with changes in certain genes, indicating that the plasmid may affect bacterial strains of limited clones exclusively. Such clones, whose fitness or virulence

is beneficial to bacteria due to a specific genetic change caused by the introduction of plasmid, become high-risk–resistant clones that are distributed worldwide. However, we did not identify special characteristics, including genetic features, of the particular clones or the mechanism behind this phenomenon. Although this was not clearly shown in a genome analysis (19), the ColE-type plasmid may affect the expression of two-component regulatory systems and type VI secretion system, which regulate the virulence, antibiotic resistance, and stress response (22, 23). In addition to the molecular mechanism, further investigation is necessary to determine whether our results can be applied to plasmids harboring other carbapenemase genes, including $bla_{KPC-2}$ and $bla_{NDM-1}$.

In this study, we investigated the change in competitive fitness and virulence in *K. pneumoniae* isolates with ColE-type plasmid bearing $bla_{OXA-232}$. *K. pneumoniae* isolates belonging to major clones such as ST11 and ST15 showed an increase in competitiveness and virulence following the introduction of $bla_{OXA-232}$-harboring plasmids, but those of minor clones did not. This may explain why certain carbapenemase-producing clones are widespread worldwide. This study revealed why the spread of antibiotic resistance by plasmids may be led by specific clones, implying that there is a point where infection control should be particularly focused.

## ACKNOWLEDGMENTS

The bacterial strains were provided by Dr. Yu Mi Wi (Samsung Changwon Hospital, Changwon, South Korea).

This research was supported in part by the Basic Science Research Program of the National Research Foundation (NRF) funded by the Korean government (MSIT) (RS-2025-00516983).

## AUTHOR AFFILIATIONS

[1]Department of Microbiology, Sungkyunkwan University School of Medicine, Suwon, South Korea
[2]Department of Biological Sciences, Sungkyunkwan University, Suwon, South Korea
[3]Division of Infectious Diseases, Samsung Changwon Hospital, Sungkyunkwan University School of Medicine, Changwon, South Korea

## AUTHOR ORCIDs

Kwan Soo Ko ⓘD http://orcid.org/0000-0002-0978-1937

## FUNDING

| Funder | Grant(s) | Author(s) |
| --- | --- | --- |
| National Research Foundation of Korea | RS-2025-00516983 | Kwan Soo Ko |

## AUTHOR CONTRIBUTIONS

Yun Young Cho, Conceptualization, Formal analysis, Investigation, Methodology, Validation, Visualization, Writing – original draft, Writing – review and editing | Hyun Woo Kim, Investigation, Writing – review and editing | Si Ho Kim, Resources, Validation, Writing – review and editing | Yu Mi Wi, Resources, Validation, Writing – review and editing | Kwan Soo Ko, Conceptualization, Funding acquisition, Project administration, Resources, Supervision, Validation, Visualization, Writing – original draft, Writing – review and editing

## ADDITIONAL FILES

The following material is available online.

## Supplemental Material

**Fig. S1 (Spectrum02126-24-s0001.pdf).** A linear map of pOXA-232 indicating the plasmid genes used in this study.

**Supplemental tables (Spectrum02126-24-s0002.pdf).** Tables S1 and S2.

## Open Peer Review

**PEER REVIEW HISTORY (review-history.pdf).** An accounting of the reviewer comments and feedback.

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
