## [Reviewer comments · Microbiology Spectrum]

Microbiology Spectrum

Different effects of plasmids harboring *bla*_{OXA-232} between major and minor clones in *Klebsiella pneumoniae*

Yun Young Cho, Hyun Woo Kim, Si-Ho Kim, Yu Mi Wi, and Kwan Soo Ko

Corresponding Author(s): Kwan Soo Ko, Sungkyunkwan University School of Medicine

Review Timeline:

Submission Date:	August 24, 2024
Editorial Decision:	December 8, 2024
Revision Received:	January 15, 2025
Accepted:	February 7, 2025

Editor: John Osei Sekyere

Reviewer(s): Disclosure of reviewer identity is with reference to reviewer comments included in decision letter(s). The following individuals involved in review of your submission have agreed to reveal their identity: Jun-Seob Kim (Reviewer #1); Fernando Hernandez Quiñones (Reviewer #2)

Transaction Report:

DOI: <https://doi.org/10.1128/spectrum.02126-24>

Re: Spectrum02126-24 (Different effects of plasmids harboring bla_{OXA-232} between major and minor clones in *Klebsiella pneumoniae*)

Dear Dr. Kwan Soo Ko:

Thank you for the privilege of reviewing your work. Below you will find my comments, instructions from the Spectrum editorial office, and the reviewer comments.

Reviewer 1 provided a very critical and damning review of the manuscript and will need to be carefully looked at before the manuscript will be done. Please ensure to address all these concerns.

Revision Guidelines

Sincerely,
John Osei Sekyere
Editor
Microbiology Spectrum

Reviewer #1 (Comments for the Author):

Please check the review file for my comments.

Reviewer #2 (Comments for the Author):

In the figures 1-3 document, figure 3 appears twice, one with KCS08 in the group of minor clones and another figure 3 with KCS033, KCS035, KCS039 isolates.

In figure 1, I would suggest adding an analysis or conclusion to the figure, within the caption, to help the general reader to interpret it.

The introduction is correct, adequately defining the problem statement and the methodology used. Likewise, the discussion manages to adequately ground the findings. It is a well done piece of work. Good luck

The manuscript titled "**Different Effects of Plasmids Harboring blaOXA-232 between Major and Minor Clones in *Klebsiella pneumoniae***" explores the impact of introducing the blaOXA-232-harboring ColE-type plasmid into various clones of *Klebsiella pneumoniae*. The study focuses on plasmid stability, competitive fitness, virulence, and gene expression. It compares the effects in major clones (ST11, ST15, ST307) and minor clones, highlighting a differential response. Major clones gain advantages in competitiveness and virulence, which could explain the global prevalence of certain carbapenemase-producing *K. pneumoniae* clones.

My comments are below.

1. Based on the genotype, major clones have multiple isolates, while minor clones have a single isolate for each genotype. Do the authors believe that this is enough to demonstrate their conclusion?

2. Authors concluded that their results explain why particular antibiotic-resistant clones are disseminated worldwide (L229). However, in this paper, only the ColE plasmid was tested, so it is importance, clar the role of the ColE plasmid in the sentence.

3. In Figure 3C, did the authors normalize the FimH expression between Major and Minor STs? If so, is there any reason that Minor STs express more FimH than Major STs?

4. There is a clear distinction between major and minor clones in terms of fitness and virulence. However, the current explanation lacks a logical basis in molecular mechanisms. The authors should provide molecular evidence to explain the physiological differences between major and minor clones after plasmid acquisition (at least hypothesis).

5. The conclusion implies that the results clarify the reasons for the global prevalence of specific carbapenem-resistant clones (such as ST11 and ST15). However, this statement might be too broad since the study only examines a limited number of clones and a single type of plasmid.

6. The study presents intriguing findings on plasmid dynamics in major and minor clones. However, it does not fully consider the clinical implications of these findings.

How do these observations affect treatment strategies or infection control policies in hospitals dealing with carbapenem-resistant *K. pneumoniae*? I hope the authors provide some insight in the discussion section.

[revised manuscript text omitted]

2022R1A2B502001716).

**Transparency declarations**

The authors declare no competing financial interests.

**References**

- **1.** Conlon-Bingham M, Hedderwick A, McKeating M, McKee M, McNally C, Lennon M,
and Aldeyab A. 2022. Preserving last resort antibiotics: A meropenem reduction strategy.
*Infect Contr Hosp Epidemiol* 43:1516-1517. <https://doi.org/10.1017/ice.2021.276>
- **2.** Codjoe, Francis S, Donkor E. 2017. Carbapenem resistance: a review. *Med Sci* 6.1:1.
<https://doi.org/10.3390/medsci6010001>
- **3.** Nordmann P, Naas T, Poirel L. 2011. Global spread of carbapenemase-producing
Enterobacteriaceae. *Emerg Infect Dis* 17:1791.
<http://dx.doi.org/10.3201/eid1710.110655>
- **4.** Potron A, Rondinaud E, Poirel L, Belmonte O, Boyer S, Camiade S, Nordmann P. 2013.
Genetic and biochemical characterisation of OXA-232, a carbapenem-hydrolysing class
D β -lactamase from Enterobacteriaceae. *Int J Antimicrob Agents* 41:325-329.
<https://doi.org/10.1016/j.ijantimicag.2012.11.007>
- **5.** Mairi A, Pantel A, Sotto A, Lavigne P, Touati A. 2018. OXA-48-like carbapenemases
producing Enterobacteriaceae in different niches. *Eur J Clin Microbiol Infect Dis*
37:587-604. <https://doi.org/10.1007/s10096-017-3112-7>
- **6.** Jeong SH, Lee KM, Lee J, Bae IK, Kim JS, Kim HS. 2015. Clonal and horizontal spread
of the *bla*_{OXA-232} gene among Enterobacteriaceae in a Korean hospital. *Diagn Microbiol*
*Infect Dis* 82:70–72. <https://doi.org/10.1016/j.diagmicrobio.2015.02.001>
- **7.** Gama J, Zilhão R, Dionisio F. 2018. Impact of plasmid interactions with the
chromosome and other plasmids on the spread of antibiotic resistance. *Plasmid* 99:82-88.
<https://doi.org/10.1016/j.plasmid.2018.09.009>
- **8.** Andrade L N, Curiao T, Ferreira J C, Longo J M, Clímaco E C, Martinez R, Coque TM.
2011. Dissemination of *bla*_{KPC-2} by the spread of *Klebsiella pneumoniae* clonal complex
258 clones (ST258, ST11, ST437) and plasmids (IncFII, IncN, IncL/M) among

[revised manuscript text omitted]

351

Per Ethics, the paper describes plasmid transfer and even provides primer sequences. If these plasmids had already been sequenced, please note that in the relevant section of materials and methods.

- The plasmid used in the paper was used in Lee et al. J. Biomed. Sci. 2020;27:1-8 (<https://doi.org/10.1186/s12929-019-0603-0>) and was cited in the paper (reference number 14). However, the plasmid number was different between original paper and the present paper, and so we matched it (pM5_OXA). Accession number (CP031737) was added, and the plasmid number was modified in the revised manuscript.

Reviewer #1

1. Based on the genotype, major clones have multiple isolates, while minor clones have a single isolate for each genotype. Do the authors believe that this is enough to demonstrate their conclusion?

- As indicated, only a single isolate for each genotype was included for minor clones, which may be limitation in this study. While many isolates were isolates for major clones, for minor clones, only a small number of isolates were collected to each genotype. So, in reality, minor clones could not contain multiple isolates. Nevertheless, we tried to supplement by including several minor clones. We mentioned it.

“While multiple isolates for each genotype could be included for major clones, only a single isolate for each genotype could be included for minor clones. It may be one of limitations in our study, but we tried to supplement the limitation by including several genotypes of minor clone.” (Line 78-81 in the revised manuscript)

2. Authors concluded that their results explain why particular antibiotic-resistant clones are disseminated worldwide (L229). However, in this paper, only the ColE plasmid was tested, so it is importance, clarify the role of the ColE plasmid in the sentence.

- As suggested, we mentioned the role of ColE-type plasmid in Discussion.

“As a representative of plasmid with blaOXA-48-like genes, it has been known that ColE-type plasmid play a role in killing other bacteria as well as antibiotic resistance (20).” (Line 234-235 in the revised manuscript)

3. In Figure 3C, did the authors normalize the FimH expression between Major and Minor STs? If so, is there any reason that Minor STs express more FimH than Major STs?

- Regrettably, we can't find any reason why minor STs express more FimH than major STs. But, one isolate (KCS33) of minor clones showed similar or less expression level of FimH, compared with isolates of major clones. We supposed that the FimH expression level would be feature of each isolate or ST. The goal of the experiment was the comparison between wild types and transconjugants, so we did not mention it.

4. There is a clear distinction between major and minor clones in terms of fitness and virulence. However, the current explanation lacks a logical basis in molecular mechanisms. The authors should provide molecular evidence to explain the physiological differences between major and minor clones after plasmid acquisition (at least hypothesis).

- As of now, the molecular mechanism explaining our results has not been elucidated. As suggested, we suggest hypothesis on the results.

“Although this was not clearly shown in a genome analysis (19), the ColE-type plasmid may

affect the expression of two-component regulatory systems and Type VI secretion system(T6SS), which regulate the virulence, antibiotic resistance, and stress response (22, 23).” (Line 248-250 in the revised manuscript)

5. The conclusion implies that the results clarify the reasons for the global prevalence of specific carbapenem-resistant clones (such as ST11 and ST15). However, this statement might be too broad since the study only examines a limited number of clones and a single type of plasmid.

- As suggested, we narrowed the scope of conclusion.

“In this study, we investigated the change in competitive fitness and virulence in *K. pneumoniae* isolates with ColE-type plasmid bearing *bla*_{OXA-232}. *K. pneumoniae* isolates belonging to major clones such as ST11 and ST15 showed an increase in competitiveness and virulence following introduction of *bla*_{OXA-232}-harboring plasmids, but those of minor clones did not.” (Line 254-258 in the revised manuscript)

6. The study presents intriguing findings on plasmid dynamics in major and minor clones. However, it does not fully consider the clinical implications of these findings. How do these observations affect treatment strategies or infection control policies in hospitals dealing with carbapenem-resistant *K. pneumoniae*? I hope the authors provide some insight in the discussion section.

- As suggested, we provide some mention the implications on the infection control.

“This may explain why certain carbapenemase-producing clones are widespread worldwide. This study revealed why the spread and spread of antibiotic resistance by plasmids may be led by specific clones, implying that there is a point where infection control should be particularly focused.” (Line 258-261 in the revised manuscript)

Reviewer #2

1. In the figures 1-3 document, figure 3 appears twice, one with KCS08 in the group of minor clones and another figure 3 with KCS033, KCS035, KCS039 isolates.

- It' our error. The first Figure 3 including KCS033, KCS035, and KCS039 isolates is right. We delete the latter figure.

2. In figure 1, I would suggest adding an analysis or conclusion to the figure, within the caption, to help the general reader to interpret it.

- As suggested, we add the conclusion of the figure in the figure legend.

“pM5_OXA was stable within a host irrespective of nutrient conditions for all transconjugants of genotype.” (Fig. 1 legend in the revised manuscript)

3. The introduction is correct, adequately defining the problem statement and the methodology used. Likewise, the discussion manages to adequately ground the findings. It is a well done piece of work. Good luck.

- Thank you.

Re: Spectrum02126-24R1 (Different effects of plasmids harboring bla_{OXA-232} between major and minor clones in *Klebsiella pneumoniae*)

Dear Dr. Kwan Soo Ko:

Please provide the data availability statement as it's required for publication

Your manuscript has been accepted, and I am forwarding it to the ASM production staff for publication. Your paper will first be checked to make sure all elements meet the technical requirements. ASM staff will contact you if anything needs to be revised before copyediting and production can begin. Otherwise, you will be notified when your proofs are ready to be viewed.

Sincerely,
John Osei Sekyere
Editor
Microbiology Spectrum

Reviewer #1 (Comments for the Author):

Authors solved all my comments.

Reviewer #2 (Comments for the Author):

read details in the attached file

Different effects of plasmids harboring bla_{OXA-232} between major and minor clones in *Klebsiella pneumoniae* review

A) In lines 258-261 of the revised manuscript it is stated: 'This study revealed why the spread and spread of antibiotic resistance by plasmids may be led by specific clones, implying that there is a point where infection control should be particularly focused'.

1) The word 'spread' seems redundant and the sentence is not fully understood, a clearer alternative is 'This may explain why certain carbapenemase-producing clones are widespread worldwide, and the major antibiotic resistance by plasmids may be led by specific clones.'

2) Stating in the conclusion that "there is a point where infection control should be particularly focused" is inaccurate unless your article addresses this, and provides arguments. Your study appears to be more oriented towards understanding the mechanisms by which there may be a greater dissemination of these plasmids and thus open the way for further research to comprehend these mechanisms in greater detail in order to establish a recommendation on the treatment and control of infections.

all the others requested suggestions have been met.